# Population genomics of *Plasmodium vivax* in Panama to assess the risk of case importation on malaria elimination

**Lucas E. Buyon**[1,2], **Ana Maria Santamaria**[3], **Angela M. Early**[1,2], **Mario Quijada**[3], **Itza Barahona**[4], **Jose Lasso**[4†], **Mario Avila**[4], **Sarah K. Volkman**[1,2,5], **Matthias Marti**[6], **Daniel E. Neafsey**[1,2]\*, **Nicanor Obaldia III**[1,3,6]\*

**1** Harvard TH Chan School of Public Health, Boston, Massachusetts, United States of America, **2** The Broad Institute, Cambridge, Massachusetts, United States of America, **3** Instituto Conmemorativo Gorgas de Estudios de la Salud, Panama City, Panama, **4** Ministerio de Salud, Panama, Panama, **5** Simmons University, College of Natural, Behavioral and Health Sciences, Boston, Massachusetts, United States of America, **6** University of Glasgow, Glasgow, United Kingdom

† Deceased.
\* neafsey@hsph.harvard.edu (DEN); nobaldia@gorgas.gob.pa (NO)

**Data Availability Statement:** The authors confirm that all data underlying the findings are fully available without restriction. Illumina sequencing

## Abstract

Malaria incidence in Panama has plateaued in recent years in spite of elimination efforts, with almost all cases caused by *Plasmodium vivax*. Notwithstanding, overall malaria prevalence remains low (fewer than 1 case per 1000 persons). We used selective whole genome amplification to sequence 59 *P. vivax* samples from Panama. The *P. vivax* samples were collected from two periods (2007–2009 and 2017–2019) to study the population structure and transmission dynamics of the parasite. Imported cases resulting from increased levels of human migration could threaten malaria elimination prospects, and four of the samples evaluated came from individuals with travel history. We explored patterns of recent common ancestry among the samples and observed that a highly genetically related lineage (termed CL1) was dominant among the samples (47 out of 59 samples with good sequencing coverage), spanning the entire period of the collection (2007–2019) and all regions of the country. We also found a second, smaller clonal lineage (termed CL2) of four parasites collected between 2017 and 2019. To explore the regional context of Panamanian *P. vivax* we conducted principal components analysis and constructed a neighbor-joining tree using these samples and samples collected worldwide from a previous study. Three of the four samples with travel history clustered with samples collected from their suspected country of origin (consistent with importation), while one appears to have been a result of local transmission. The small number of Panamanian *P. vivax* samples not belonging to either CL1 or CL2 clustered with samples collected from Colombia, suggesting they represent the genetically similar ancestral *P. vivax* population in Panama or were recently imported from Colombia. The low diversity we observe in Panama indicates that this parasite population has been previously subject to a severe bottleneck and may be eligible for elimination. Additionally, while we confirmed that *P. vivax* is imported to Panama from diverse geographic locations, the lack of impact from imported cases on the overall parasite population genomic profile

reads are available through the NCBI Sequence Read Archive Bioproject: PRJNA655141. Accession numbers:SRX8882313-SRX8882371.

**Funding:** This project has been funded in whole or in part with Federal funds from the National Institute of Allergy and Infectious Diseases, National Institutes of Health, Department of Health and Human Services (U19AI110818) to the Broad Institute. Authors funded: D.E.N. This study was supported in part by core funds from the Gorgas Memorial Institute of Health Studies of Panamá and a grant contract from the National Secretary of Science, Technology, and Innovation (SENACYT) of Panama: grant contract SENACYT-FID16-P-097; the Harvard TH Chan School of Public Health, Boston, MA. USA; the Ministry of Health of Panama and the Sistema Nacional de Investigación (SNI) of Panamá. Authors funded: N.O. The funders had no role in study design, data collection and analysis, decision to publish, or preparation of the manuscript.

**Competing interests:** The authors have declared that no competing interests exist. Author Jose Lasso was unable to confirm their authorship contributions. On their behalf, the corresponding author has reported their contributions to the best of their knowledge

suggests that onward transmission from such cases is limited and that imported cases may not presently pose a major barrier to elimination.

## Author summary

*Plasmodium vivax* is a major global health threat particularly in Central and South America which experiences 700,000 *P. vivax* cases each year. Panama has greatly reduced *P. vivax* incidence, however, this progress has since plateaued. Understanding how the parasite moves throughout the country, uncovering pockets of focalized transmission, and identifying imported cases, is critical for Panama and other countries to succeed in their elimination efforts. Genomic epidemiology and population genomics tools can help provide this information needed to inform malaria control policy. In this study, we collected 100 Panamanian *P. vivax* samples from two time periods (2007–2009 and 2017–2019), of which 59 yielded usable sequencing data. We found that the majority (n = 47) samples belong to a single highly related lineage, termed CL1. This lineage has persisted since at least 2007. We also observed a second smaller completely clonal lineage of four parasites, termed CL2. Additionally, we observed four samples that shared no recent ancestry with any other Panamanian samples but clustered with samples collected in a previous study from Colombia. We highlight how genomic epidemiology can be used to spotlight parasites that may be imported as a result of human migration, as well as corroborate or refute the country of origin as suggested by the travel history of a patient. There is no evidence of outcrossing between these potentially imported parasites and the local Panamanian parasite population. This finding suggests that imported parasites are not driving ongoing malaria transmission in Panama. We note the need for sustained genomic surveillance of *P. vivax* in Panama to monitor transmission dynamics in the local population and to further flag potentially imported cases. The low diversity we observe in Panama indicates that this parasite population has been previously subject to a severe bottleneck and may be eligible for elimination.

## Introduction

Malaria is a parasitic disease transmitted by the bite of female *Anopheles* mosquitoes. Malaria parasites cause approximately 219 million cases and 435,000 deaths each year, the vast majority in sub-Saharan Africa [1]. *Plasmodium falciparum*, the most virulent of the six *Plasmodium* species that infect humans (*P. falciparum, P. vivax, P. malariae, P. ovale wallikeri, P. ovale curtisi, and P. knowlesi*), causes the majority of these cases [1]. Though billions of dollars have been devoted to the control and eradication of malaria caused by *P. falciparum*, comparatively little attention has beens given to *P. vivax*, the most prevalent malaria parasite outside Africa [2]. The impact of *P. vivax* on human health was once considered minimal, relative to the more virulent *P. falciparum* [1,2]. However, recent studies suggest *P. vivax* causes a significant global health burden [1,2]. The *P. vivax* lifecycle includes a dormant liver "hypnozoite" stage [2].The hypnozoite stage can cause a relapse of malaria weeks to months after the initial infection, thus beginning the cycle of infection and complicating control efforts [2].

Sixty percent of the Central and South American population lives in areas with ongoing malaria transmission, predominantly caused by *P. vivax* [3]. The region experiences about 700,000 *P. vivax* cases each year [1]. Between 2000 and 2015 the incidence rate of malaria fell

37% globally and 42% in Africa [1]. In the Americas during the same period, malaria mortality decreased by 72% [1]. Unfortunately, recent evidence suggests that this trend has stalled, and in some countries malaria incidence has even increased [1]. Panama eliminated the autochthonous transmission of *P. falciparum* in 2010, outside of a small outbreak on the Colombian border in 2015 [4]. Since 2010, *P. vivax* has caused almost all malaria cases in Panama [1,5,6]. *P. vivax* cases in Panama have declined precipitously since 2005, from close to 1 case per 1000 persons, to under 0.25 cases per 1000 persons in 2017 [7]. However, malaria incidence in Panama has plateaued since 2008. This plateau in incidence could be due to low levels of transmission and/or imported cases that are re-seeding infections.

Human movement leading to parasite migration is a potentially significant epidemiological threat to malaria control in Panama. Parasite importation stemming from human migration is a challenge to elimination programs in other countries around the world [8–10]. The unique geographic position of Panama makes it a crossroads for human migration to the United States from South America [5,11]. Migrants enter Panama through two paths: through the Darien jungle region on the border with Colombia, and through the Kuna Yala Amerindian reserve ('Comarca') on the Caribbean coast [5,6,11]. Previous studies implicate these regions as focal points of ongoing malaria transmission in Panama and suggest this is partly due to imported parasites [5,11]. It is estimated that approximately 60,000 continental and extra-continental migrants crossed the southern border of Panama through the Darien jungle region in 2015 and 2016 [12].

To inform effective malaria elimination strategies in Panama, it is critical to understand how the parasite moves throughout the country, uncover pockets of focalized transmission, and differentiate between sustained local infection and case importation as the reason for disease persistence. [4,5]. Whole-genome sequencing can help paint a detailed picture of parasite movement and transmission within and between countries [10,12,13]. However, *P. vivax* cannot be grown *in vitro*, and the difficulty of sequencing *P. vivax* from clinical samples dominated by host DNA has hindered parasite population studies [14]. Recent advances such as hybrid capture [15] and selective whole genome amplification (SWGA) mitigate this problem by allowing for parasite DNA to be selectively enriched before sequencing [14]. Both methods have allowed for population genomic studies of *P. vivax* using samples directly from patients.

In this study, we describe the population genomics of *P. vivax* in Panama over a 12-year time span, with the aim of understanding patterns of genetic variation and recent shared ancestry (relatedness) at different geographical and temporal scales. We found the vast majority of *P. vivax* cases in Panama belong to a single highly related lineage that has persisted for at least a decade. Furthermore, we observed a second smaller clonal lineage concentrated near the Panamanian-Colombian border. We also found several samples that shared no relatedness with any other sample, which may represent either localized pockets of outbred *P. vivax* transmission or imported cases. Revealing these patterns of relatedness among parasite infections can help inform best strategies for targeting interventions or case investigation methods to increase the likelihood of successful elimination. We discuss these findings and their implications for ongoing elimination efforts of *P. vivax* in Panama. The results obtained from this study will help inform future elimination efforts in Panama and the rest of Meso-America.

## Materials and methods

### Ethics statement

The Research Bioethics Committee (CBI) of the Gorgas Memorial Institute of Health Studies gave the study ethical approval (Permit: 154/CBI/ICGES/17). Written consent was obtained from infected patients prior to collecting samples.

## Sample collection

We collected 96 *P. vivax* samples from infected consenting volunteers identified through passive or active surveillance by technicians from the Department of Vector Control, Ministry of Health (MINSA) of Panama. Two groups of DNA samples from infected patients were used in this study: 1) 56 DNA samples collected during 2007–2009 and 2) 40 DNA samples collected during 2017–2019. The 2007–2009 samples were collected as part of an earlier study exploring the genetic diversity of *P. falciparum* and *P. vivax* in Panama (Approved by The National Committee for Research Bioethics of Panama (CNBI): Permit: 468/CNBI/ICGES/06, PI: José E. Calzada). The Gorgas Memorial Institutional Animal Use and Care Committee (CIUCAL) (Permit: 002/CIUCAL-ICCES-2012) approved the use of Aotus *P. vivax* AMRU-1 and SAL-1 infected blood samples as a source of control DNA. Patient blood samples were collected by finger-prick with a lancet and spotted into EBF 903 Five Spot Blood Cards (Eastern Business Forms, INC, SC, USA). The samples were then transported at ambient temperature to the laboratory and stored at –20˚C until processing. Thin and thick blood smears were obtained from patient samples. The blood smears were stained with Giemsa for percent parasite density determination, species identification, and stage differential counts. Each volunteer donated ~150 μL of blood.

## Information survey

We collected demographic, geographic, socioeconomic, and epidemiological information from each study subject using an epidemiological form developed for the Survey123 for ArcGIS online survey program (Esri, Redlands, CA), as allowed under ethical approval.

## Malaria microscopy

Giemsa stained thick and thin blood smears were examined by light microscopy for parasite density determinations, P*lasmodium* species confirmation, and parasite lifecycle stage count. Parasite densities were calculated by quantifying the number of malaria-infected red blood cells (iRBCs) among 500–2000 RBCs on a thin blood smear and expressing the result as percent parasitemia (percent parasitemia = parasitized RBCs /total RBCs) x 100).

## DNA extraction

We extracted DNA from the filter paper blood spots using the Chelex method as described [16] for samples obtained during 2007–2009 and with the Qiagen DNA mini kit for samples obtained during 2017–2019.

## Molecular confirmation of *P. vivax* infection

We confirmed *P. vivax* infection for all samples collected during 2017–2019 by amplification of the *P. vivax* PVX_18SrRNA gene using a qRT-PCR assay as described [17].

## Selective whole genome amplification and sequencing

We carried out DNA pre-amplification as described [14]. Briefly, the thermocycler was preheated to 35˚C. We dispensed aliquots of 37μl of Power SYBR Green Master Mix, plus 3μl phi39 into each PCR tube, next adding DNA and water to achieve a final volume of 47μl. Thermocycler settings were as follows: 35˚C x 10 min; 34˚C x 10 min; 33˚C x 10 min; 32˚C x 10 min; 31˚C x 10 min; 30˚C x 16 hours; 65˚C x 10 min; and, 4˚C for infinity. SWGA reaction products were diluted with 50 μl of water. We purified 50 μl of the diluted product using 50 μl AmPURE beads according to the instructions of the manufacturer. We then eluted beads in

30 μl of water. Approximately 60–120 ng/μl of DNA was obtained after bead purification of the SWGA reaction. We measured DNA concentration using Nanodrop quantitation.

## Whole-genome sequencing

We performed whole-genome sequencing (WGS) on all 96 *P. vivax* samples using Nextera libraries and an Illumina HiSeq X platform. Sample reads were aligned to the P01 reference genome assembly using BWA-MEM, version 0.7 [18].

## SNP discovery and quality filtering

We marked duplicate reads using the MarkDuplicates tool from Picard tools. We next performed local realignment around indels using the Genome Analysis Toolkit (GATK) RealignerTargetCreator and GATK IndelRealigner (GATK Version 3.5.0). We called variants using GATK HaplotypeCaller using best practices to call and filter single nucleotide polymporphisms (SNPs) and generate individual variant call files (gVCFs) s for each sample. We called variants in two batches, one containing samples collected in 2007–2009 and one containing samples collected in 2017–2019. We performed joint variant calling on the sets separately using GATK GenotypeGVCFs tool with GATK hard filters, including calls in subtelomeric regions. The resulting datasets consisted of 56 samples and 407,554 sites for the 2007–2009 samples, and 40 samples and 171,433 variants for the 2017–2019 samples. We retained samples for analysis if they exhibited a minimum mean read depth of five and had calls at more than 80% of variant sites in the dataset corresponding to their collection period, including those in subtelomeric regions. We calculated and evaluated data quality measures using the VCFtools package and custom R scripts [19]. Thirty-five samples from 2007–2009 and 24 samples from 2017–2019 passed these filters and were kept for further analysis.

We next used GenotypeGVCFs tool to construct a joint dataset with the 59 Panamanian samples plus a collection of previously collected global samples (Bioproject numbers PRJNA240356-PRJNA240533 [20]). The joint dataset contained 168 samples and 2,250,245 variants. We filtered sites on the basis of quality (GQ > 40), passing VQSR truth sensitivity level of 0.99 or greater, missing rate (having a call at that site in > 85% of samples). We also excluded any sites that were not bi-allelic and indels. The joint dataset generated after filtering contained 168 samples and 62,211 sites.

Lastly, we generated a dataset containing SNPs found jointly in 80% of both the 2007–2009 and 2017–2019 samples. We also filtered sites in this dataset by excluding non-biallelic SNPs and on the basis of quality (GQ > 30), and passing GATK filters. The resultant dataset contained 56 samples and 2,335 SNPs for the 2007–2009 samples, and 40 samples and 1,301 SNPs for the 2017–2019 samples. For these samples, we generated a highly filtered variant set containing biallelic SNPs that passed the GATK filters (GQ > 30, truth sensitivity level > 0.99, Mean DP > five) and were called in at least 80% of the samples from both time periods (2007–2009; 2017–2019). Calls from the two sample sets were merged to create a unified dataset of 96 samples and 264 genotyped SNPs.

## Determination of sample clonality

We estimated sample clonality using the $F_{ws}$ statistic. $F_{ws}$ measures the within-sample genetic diversity (measured by heterozygosity $H_w$) relative to the overall population genetic diversity ($H_s$) [21]. The underlying theory assumes that a monoclonal (single strain) infection has extremely low genetic diversity relative to overall population genetic diversity. By contrast, a polyclonal (multiple strain) infection has high diversity relative to overall population diversity (compared to a monoclonal infection). By estimating the ratio between within-host diversity

and population diversity, we can distinguish between monoclonal and polyclonal infections [21]. A sample with an $F_{ws}$ statistic of 0.95 or greater ($> = 0.95$) is considered monoclonal. We calculated $F_{ws}$ using the R package moimix [22].

### Analysis of recent common ancestry

We used hmmIBD [23] to estimate the proportion of sites identical by descent (IBD) between sample pairs to ascertain recent common ancestry among Panamanian and Colombian samples collected previously from a global *P. vivax* population study [20]. We estimated minor allele frequency (MAF) for IBD inference using the genetically distinct Panamanian samples, a representative sample from each of the two highly related Panamanian clusters, and the Colombian samples. We included Colombian samples to improve MAF estimation given the greater genetic diversity of the Colombian parasite population and presumed historical gene flow with Panama. We subsetted the master dataset file to keep only samples collected in Panama and Colombia. Sites were excluded sites on the basis of minimum and maximum read depth (five and thirty respectively) to ensure that we were using only high-quality SNPs. The input dataset for hmmIBD contained 89 samples (59 Panamanian samples and 30 Colombian samples) and 15,788 variant sites. We then re-formatted the data using a custom perl script for input into hmmIBD along with the MAF estimates. We conducted analysis and visualization of the hmmIBD output using custom R scripts.

### Analysis of population structure

We employed principal components analysis (PCA) and a neighbor-joining tree to study the population structure of Panamanian samples in the context of the worldwide *P. vivax* population [20]. We used a strictly filtered SNP set for PCA, keeping only variants with calls in at least 95% of samples. This input dataset consisted of 168 samples and 2,428 variants. We used the R package SNPRelate to conduct PCA [24]. Covariation within the two clusters heavily influenced the PCA of all samples, so we also performed PCA using a single consensus sequence for each cluster.

We used the R packages ape, StAMPP, pegas, and adegenet, [25–28] to generate the neighbor-joining tree and genetic distance statistics. First, we calculated Nei's distance for all pairwise sample combinations using the master dataset consisting of 168 samples and 62,211 sites to generate a distance matrix. The distance matrix was used to generate a tree. We used the bootphylo function in the ape package to bootstrap the dataset 100 times to estimate nodal support. We then visualized the final tree with support values using the FigTree program [29]. We used R software (R version 3.6.1) to carry out statistical analysis and data visualization.

## Results

### Recent common ancestry analysis reveals single highly related lineage of parasites

We successfully generated usable sequencing data from 35/56 (58%) Panamanian *P. vivax* samples collected between 2007–2009 and 24/40 (60%) collected between 2017–2019, for a total of 59 samples (Fig 1A). All Panamanian samples had an $F_{ws}$ statistic greater than 0.95, indicating that they were all monoclonal (Fig 1B).

We next analyzed the 59 Panamanian genomes in the context of 109 previously published *P. vivax* genomes, generating a filtered dataset consisting of 168 samples and 62,211 high-quality biallelic SNPs.

We used hmmIBD [23] to estimate the proportion of the genome that is identical by descent (IBD) among Panamanian sample pairs to understand patterns of recent common

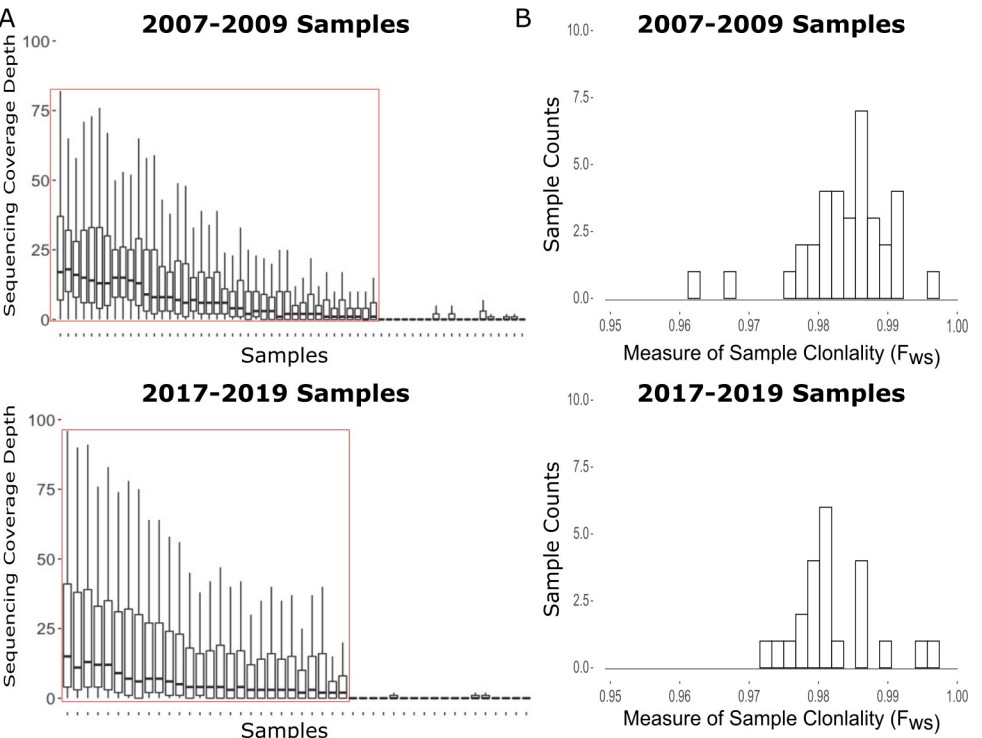

**Fig 1. Sequencing and Sample Assessment at Variant Sites.** A) Distribution of variant site read coverage for each sample stratified by the collection period. Coverage values > 100 were censored for visualization purposes. Samples within the red boxes were kept for analysis. B) Distribution of $F_{ws}$ values for all samples, stratified by the collection period. We interpreted $F_{ws}$ values > 0.95 as evidence of sample monoclonality.

ancestry. IBD measures the proportion of the genome between two individuals that was inherited from a recent common ancestor. Pairwise IBD values closer to 100% indicate very recent common ancestry. We subsetted the dataset to contain only samples collected in Panama and Colombia to estimate pairwise IBD. We strictly filtered sites on the basis of minimum and maximum read depth (five and thirty respectively), resulting in a dataset with 89 samples and 15,788 sites for input into hmmIBD.

We observed a bimodal distribution of pairwise IBD in Panamanian samples, with peaks near zero and 0.95 (Fig 2A). Forty-seven of the 59 Panamanian samples shared high IBD (>0.875) with each other, indicating very recent common ancestry. Four other Panamanian samples, all collected in the Kuna Yala Province, shared 100% IBD with each other, and 0–10% IBD with any other sample, Panamanian or Colombian (Fig 2A and 2B). Another four Panamanian samples exhibited no IBD with each other nor any of the other Panamanian samples. All four of these samples were collected in the Darien jungle region or Kuna Yala, which are the two main points of entry for migrants traveling through Panama. These four samples drive the modal peak of pairwise IBD at zero.

The variable degree of relatedness among the 47 samples sharing > 0.85 IBD suggested that data quality potentially impacted the estimation of IBD. We plotted the relationship between IBD and sample quality, measured by the average proportion of high coverage sites in each sample pair, to determine if pairwise sample data quality affected the estimation of IBD (S1 Fig). We defined high coverage sites as sites with greater than 5x coverage (the cutoff for site filtering). We observed that as the average proportion of high coverage sites for sample pairs increased, pairwise IBD estimates correspondingly increased as well (S1 Fig). This relationship

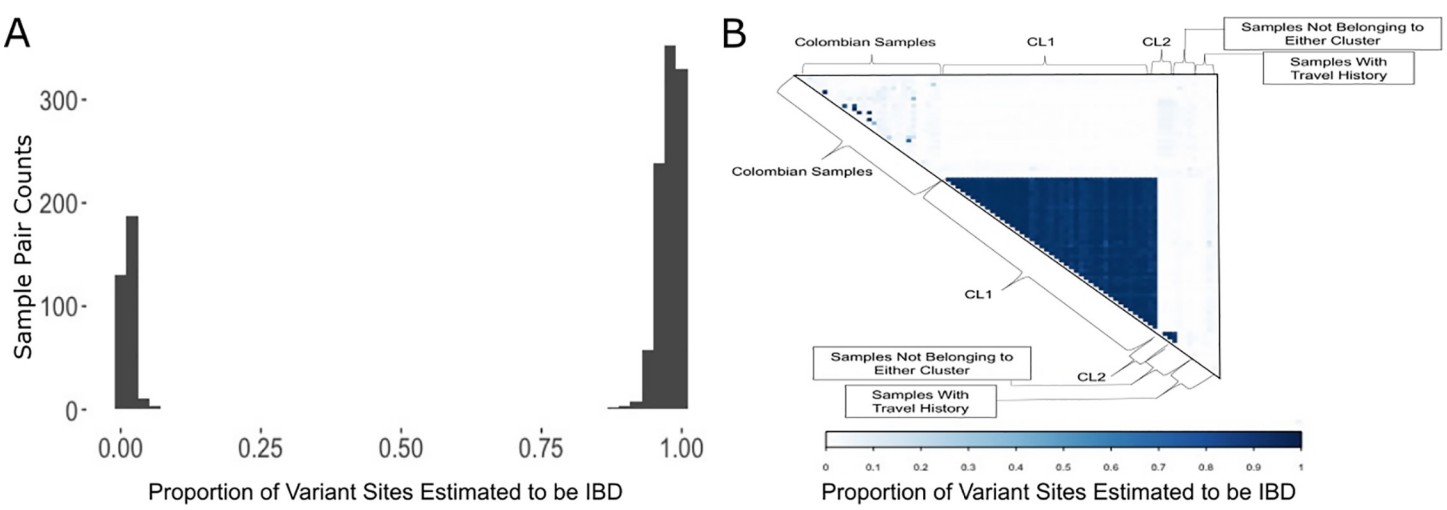

**Fig 2. IBD analysis of the Panamanian samples.** A) The distribution of pairwise IBD estimates among the Panamanian samples. IBD values near zero indicate no recent common ancestry. Values closer to one indicate that the sample pair are clonal or essentially clonal. B) Depicts heatmap of pairwise IBD values for Panamanian and Colombian samples.

suggests that poor data quality can lead to underestimation of IBD. It is possible that the majority of the pairwise IBD estimations would be closer to one had the overall sample sequence quality been higher. The prevalent highly genetically related lineage is referred to henceforth as cluster one (CL1). CL1 samples share an IBD fraction of at least 0.875 with other samples in this cluster. We also concluded that the four samples that shared 100% IBD with each other constituted a second completely clonal lineage, henceforth referred to as cluster two (CL2).

Next, we examined how these two clusters and the other Panamanian samples not belonging to either lineage were geographically distributed in Panama (Fig 3). Samples from CL1 were found across Panama. Notably, samples collected from both 2007–2009 and 2017–2019 were found in this lineage. The inclusion of samples from both collection periods demonstrates that this lineage has persisted throughout Panama for at least a decade. We did not find any evidence of structure in the *P. vivax* population by region or relative to the Panama Canal, as was previously observed for *P. falciparum* [4].

We only observed samples belonging to CL2 in a specific locality, Puerto Obaldia, in the Kuna Yala Amerindian territory (Comarca) along the Atlantic Coast. We lacked geographic information for one of the four samples in CL2. Additionally, of four samples that shared no recent common ancestry with any sample in the dataset, three were collected in Darien province, along the Colombian border and one was collected in Kuna Yala.

After identifying two highly related lineages in Panama, we explored an approach for determining whether the samples excluded from analysis due to low coverage could belong to one of these lineages. We identified a set of 264 genotyped SNPs that were called in at least 80% of samples across both Panama sample collection periods. We then calculated Nei's standard genetic distance on all pairwise sample comparisons. The majority of excluded samples across both collection periods (17/21 and 4/16 for the 2007–2009 and 2017–2019 collection periods respectively; a total of 21/37 samples) exhibited very low levels (0–1%) of genetic distance with the CL1 samples and higher genetic distance (0.2–0.25) with the CL2 samples (S2 Fig). Seven samples in the 2017–2019 collection period had a high proportion of missing calls for these 264 SNPs, making distance measures uninformative. Three excluded samples from 2007–2009

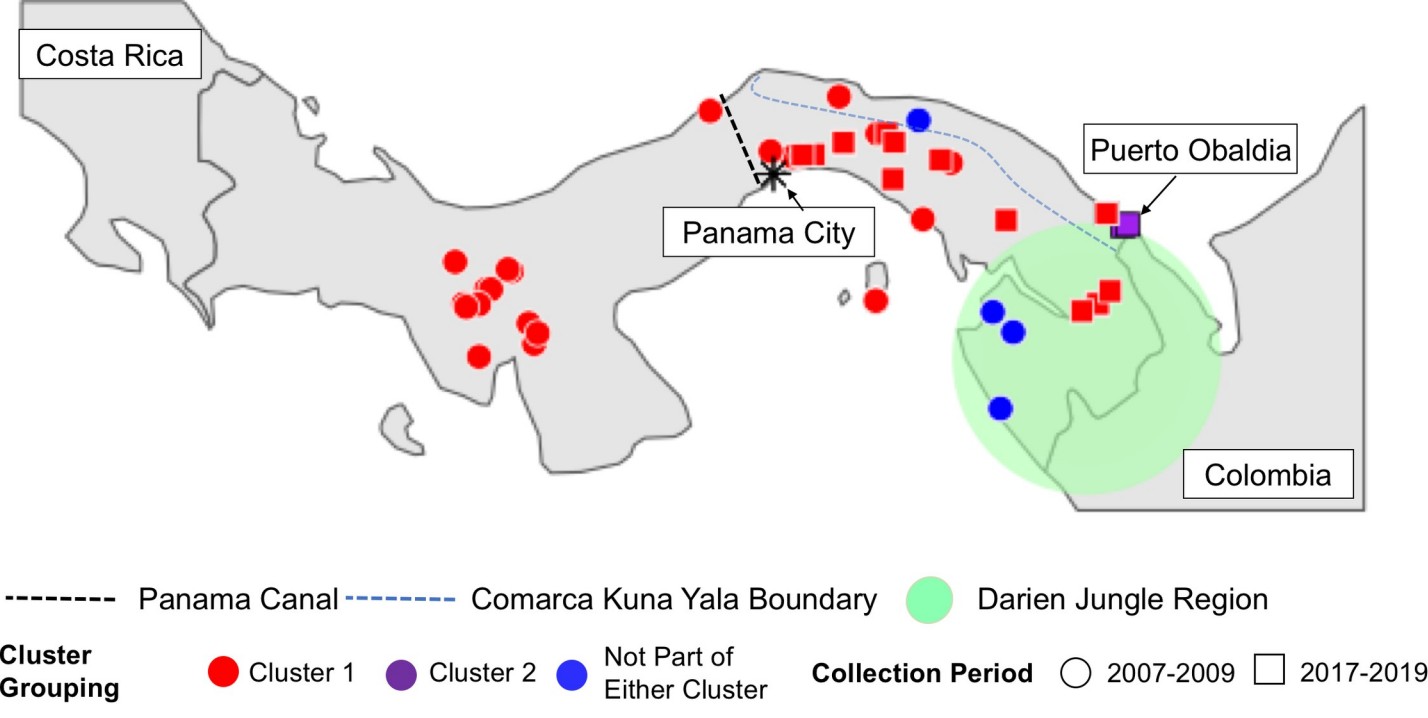

**Fig 3. Map of Panamanian sample collection sites.** Sample colors show which cluster (or neither) each sample belongs too. Shape indicates the sample collection period. The dotted line shows the location of the Panama Canal. The Blue Line shows the border of the Comarca Kuna Yala. The Darien Jungle Region is indicated by the green shaded area.

collected in the Darien Jungle Region had relatively high genetic distance from all other samples in the dataset. Two samples from the 2017–2019 collection period exhibit very low (0–0.1) genetic distance with the CL2 samples, and higher (0.15–0.25) genetic distance with the CL1 samples. The previously observed sample clustering patterns did not change when conducting the analysis with the 264 SNPs.

### Exploring the regional context of Panamanian *P. vivax*

We built a neighbor-joining tree using the Panamanian samples plus previously sequenced samples [20] (Fig 4A) to understand the Panamanian *P. vivax* population in a global context. As noted in previous studies [20,30] we observed clusters of samples corresponding to different geographic regions, with a large cluster of Central and South American samples. CL1 and CL2 formed distinct clusters within the Central and South American cluster with 100% bootstrap support. CL2 is situated in a cluster containing samples from Colombia, with 100% bootstrap support at deep nodes. While these four samples are clustered together with 100% support and exhibit short branch length, a long branch connects them to the rest of the Colombian cluster. The Panamanian samples that shared little IBD with either cluster also grouped with the Colombian samples. These samples appear to share distant ancestry with each other and the rest of the Colombian samples. The samples also formed their own sub-cluster within the Colombian cluster.

PCA conducted with worldwide samples showed tight clustering of all Central and South American samples, with only one Panamanian sample falling outside this Central and South American cluster (S3 Fig). PCA restricted to the samples collected from Central and South America is heavily influenced by covariation among samples within the two clusters (S4 Fig). PCA performed with a single consensus sequence representing each cluster revealed CL1

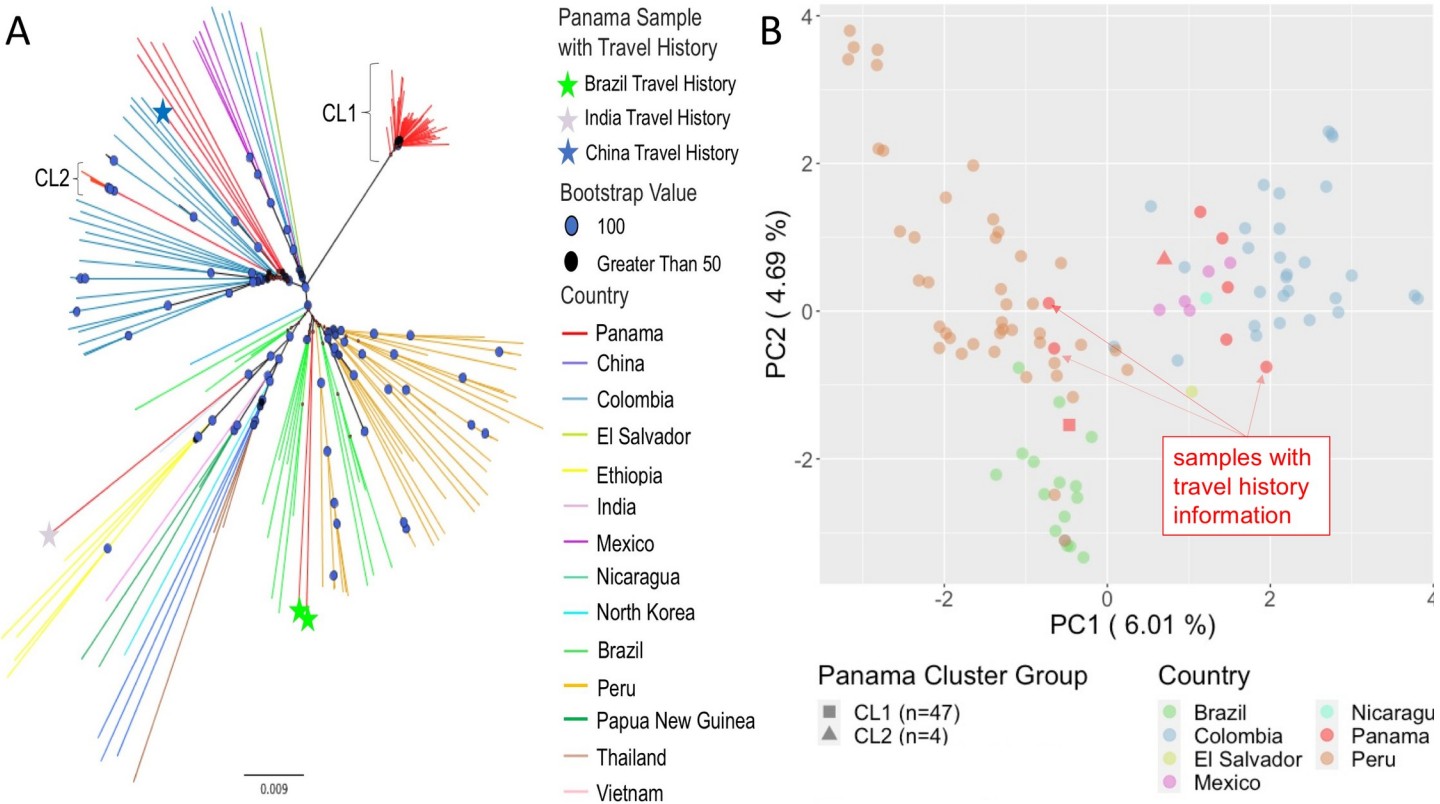

**Fig 4. Population Structure.** A) Neighbor-joining tree for all samples worldwide. Node symbols denote support values: circles indicate 100% support, triangles indicate > 50% support. Branch color indicates the country of collection for each sample. Panamanian samples with travel history are noted with the colored stars. B) PCA of Central and South American samples. Circle color indicates country of collection. Consensus sequences for cluster one and cluster two are noted as a square and triangle respectively. Panamanian samples with travel history are annotated.

clusters with samples from Peru and Brazil and CL2 clusters with the Colombian samples (Fig 4B). All four outbred Panamanian samples that shared no recent ancestry with the other samples also clustered with the Colombian samples. Principal component one differentiated CL1 and samples from Brazil and Peru from CL2 and the rest of the Central and South American samples.

## Genomic data are concordant with travel history in three out of four cases

Four of the 59 samples had travel history data associated with them. All of these samples with travel history data were collected during the 2007–2009 period. Travel history information suggested that two samples were originally from Brazil, one sample from India, and one sample from China. The two samples with Brazilian travel history fell within the Brazilian cluster on the NJ tree and clustered with Brazilian samples on PCA (Fig 2A and 2B). The one sample with Indian travel history grouped with the other Indian samples on the NJ tree, and clustered with the other Indian samples via PCA as well (Figs 2A and S4). This sample was the only one collected in Panama to fall outside of the Central and South American cluster in the PCA with the worldwide sample set. For the two samples with Brazilian travel history and one sample with Indian travel history, genomic data supported the same country of origin as the travel history information.

The sample with Chinese travel history had a discrepancy between the region of origin suggested by its travel history information and its genomic data. This sample clusters with the

Central and South American samples on the worldwide PCA instead of with the samples from China. This sample clustered with the Colombian samples in the PCA conducted with only the Central and South American samples (Fig 4B). Similarly, on the NJ tree, this sample fell within the Colombian cluster with 100% bootstrap support along with the four Panamanian samples that shared zero IBD with other Panamanian samples in the dataset.

## Discussion

Panama is on the cusp of eliminating malaria after several decades of intervention [5]. We found extremely high clonality in the Panamanian *P. vivax* population, observing that the majority of the successfully sequenced samples (47/59) belonged to a single highly related lineage, CL1. CL1 has persisted throughout Panama for at least a decade, in spite of ongoing elimination efforts. Sample contamination could not explain this pattern as samples were collected in two collection periods 10 years apart and extracted, amplified, and sequenced separately. Our study suggests that the Panamanian *P. vivax* population has been through a strong bottleneck due to reduced transmission, resulting in the majority of the population belonging to a single highly related lineage. Similar reductions in clonal diversity of *P. vivax* populations have been observed elsewhere. For example, a study investigating the relationship between *P. vivax* transmission intensity and genetic diversity in Malaysia [31] documented that when there is a decline in parasite transmission, there is an increase in the clonal composition of the population. Several studies of *P. falciparum* genetic diversity and transmission intensity from Senegal [32], Thailand [33,34], and Colombia [34] have also noted the same relationship. Furthermore, there is evidence of persistence and transmission of *P. vivax* clonal lineages in Malaysia [31], and *P. falciparum* clonal lineages in Colombia [35], Ecuador [36], and Haiti [37]. Our study demonstrates a similar relationship in Panama between low transmission and extremely low genetic diversity of the *P. vivax* population. Almost all CL1 samples (46/47) share IBD > 0.95 with at least one other CL1 sample, suggesting a substantial fraction of this population is clonal. Several previous studies found that the Central and South American *P. vivax* populations are distinct from each other [20,30,38]. A previous study suggested this population structure is due to multiple founding events after likely European introduction [39]. This structure could also be due to genetic drift since founding. However, the Panamanian *P. vivax* population has been through too severe of a bottleneck to help clarify historical causes of this population structure with the present data. Both scenarios point to the need for further longitudinal genomic studies of *Plasmodium* parasites to better understand population dynamics over space and time.

Previous studies have indicated that Panama has focal transmission in indigenous regions (Comarcas) [5,6,11]. Malaria transmission in Panama is increasingly concentrated in the Comarcas, with the proportion of total malaria cases in Panama reported from the Comarcas rising from 41.8% in 2005 to 90% in 2019 [40]. Prior work shows that low transmission can lead to an increase in clonal population structure [33]. The finding that CL1 is distributed ubiquitously throughout Panama is unexpected given the concentration of the malaria epidemic within spatially separated regions of the country. The geographic distribution of CL1 suggests that parasites have historically moved throughout the country, founding new populations or supplanting small existing ones. Case investigations and understanding human movement patterns throughout Panama will be critical to achieving elimination.

This study had some limitations. We were unable to generate high-quality sequencing data from ~40% of the samples. Factors such as differences in DNA extraction techniques used for the two sample collection periods or length of storage of the samples could have affected DNA yield and/or molecular weight, impacting SWGA. Some samples may have had lower coverage

due to lower parasitemia. Dissimilarities in coverage between the early and late sample batches could be due to technical factors such as different flowcell loading. We did not find an association between coverage and geographic location. Low sequencing coverage for some samples may have limited the sensitivity of the $F_{ws}$ statistic to detect polyclonal infections. However we used a filtered dataset of SNPs for $F_{ws}$ calculation that had a minimum of coverage of 5x and a maximum coverage of 30x to minimize bias from coverage. We also used a small set of 264 SNPs that were called in ~80% of samples to calculate Nei's standard genetic distance to determine if the excluded samples were genetically distant from CL1 or CL2. We found that the majority of the excluded samples were genetically similar to CL1 (S2 Fig). This result indicates that our assessment of relatedness within the Panamanian *P. vivax* population is not biased by parasitemia or other factors that could have affected sequencing success. This finding also suggests that we did not miss additional genetically distinct circulating Panamanian *P. vivax* strains and thus did not bias our analysis by excluding these samples. Additionally, all samples that did yield usable sequence data were distributed across almost all localities across Panama. The exception to this was a group of samples collected near the Panamanian-Costa-Rican border from which we were unable to generate usable sequence data. However, most ongoing malaria transmission in Panama occurs East of the Panama Canal, where most of the samples that generated usable sequencing data were collected [6,11]. Due to the geographic sampling coverage of regions with ongoing malaria transmission, we believe these data are reflective of the current state of the Panamanian *P. vivax* population. We also lacked geographic collection data for two of the successfully sequenced samples, and they were excluded from the geographic analysis. The lack of geographical data is unlikely to bias our conclusions since these samples came from both different collection periods and regions. The two samples also constitute a small proportion of samples in the final dataset.

Genomic epidemiology can help to support malaria elimination efforts in Panama in multiple ways. First, genomic data can help identify genetically distinct cases that may be imported. Panama sits at the crux of migration paths to the United States, and it is possible that genetically distinct samples collected in Panama represent imported cases. Integrating travel history information with genomic data can help solidify the identification of imported cases. Four samples had patient travel history information, and genomic data supported the presumed country of origin for three of them. The fourth sample was collected from a subject with travel history from China. However, it clustered with Colombian samples on the NJ tree and the PCA, suggesting the infection was likely acquired somewhere in Central or South America, rather than China. Relapsing *P. vivax* infections resulting from dormant hypnozoites could complicate reconciliation of travel history with genomic data if infections were acquired months previously. Further development of tools using a benchmarked set of markers, such as SNP barcodes [41,42], for each *P. vivax* endemic country would help to identify parasite country of origin solely using genomic data.

Second, genomic data will be critical to determine if imported parasites are contributing to local transmission and/or admixing with the local parasite population. For example, we did not observe evidence of admixture between the imported samples from India and Brazil and the samples that comprise CL1, or evidence of onward clonal transmission of the imported samples. Our data cannot distinguish whether CL2 is a native Panamanian parasite lineage or if it has been imported from Colombia. However, the four CL2 samples displayed a genomic and epidemiological pattern consistent with recent local transmission, as all CL2 samples are virtually identical and were collected from the same municipality in 2019.

Overall, the existence of one main parasite genetic lineage exhibiting no recent evidence of outcrossing with imported infections suggests that Panama is ripe for the elimination of *P. vivax*. While case importation remains a threat, the lack of evidence of outcrossing suggests it

may not be sufficient to prevent elimination under present circumstances. The potential for genomic data to identify imported cases in Panama will be improved by collecting genomic data from other countries in the region as a population genomics reference. Ongoing genomic surveillance paired with case containment efforts will also be needed to mitigate the risk of outbreaks resulting from imported cases and prevent reversal of the impressive progress that has been recently made towards malaria elimination in Panama.

## Supporting information

**S1 Fig. Pairwise IBD Estimates Increase with Sample Quality.** Depicts pairwise IBD estimates for all Panamanian sample pairs with IBD > 0.875 plotted against the mean proportion of high coverage sites (sites with > 5x coverage) in each sample pair. The line indicates a linear regression, the box displays the Pearson correlation coefficient between the two axes variables. (PNG)

**S2 Fig. Annotated heatmap of pairwise Nei's standard distance comparisons between all 2007–2009 and 2017–2019 samples using SNPs that were callable in at least 80% of samples. Each block row and column presents a single sample**. Brackets indicate sample groups. (PNG)

**S3 Fig. Principal components analysis of Panama samples and previously collected samples from Central and South America, Asia, and Africa.** Samples are colored by the region of origin. Parentheses contain the percentage of variance explained by each principal component. (PNG)

**S4 Fig. Principal components analysis of Panama samples and previously collected Central and South American samples.** Samples are colored by country of origin. Parentheses contain the percentage of variance explained by each principal component. (PNG)

## Acknowledgments

The authors thank José Calzada for generous contribution of the 2007–2009 samples and for support of this study. We also thank Selina Bopp and Becky Kuzma at HSPH for conducting SWGA of the *P. vivax* samples. We also thank at ICGES, Néstor Sosa Azael Saldaña and Gladys Tuñón, for their administrative support; also at ICGES Marlon Núñez, Ariel Magallón, and José C. Marín for their technical and logistic support; at the University of California-San Diego, Elizabeth Winzeler and Annie Cowell for their technical advice on establishing SWGA for *P. vivax*. Department of Research and Development (I+D), Secretary of Science, Technology, and Innovation (SENACYT) of the Government of the Republic of Panamá, Iriela Aguilar, and Milagros Mainieri for their administrative and technical support with project management. We dedicate this manuscript to our co-author Jose Lasso, who sadly passed away during the drafting of the manuscript.

## Author Contributions

**Conceptualization:** Itza Barahona, Sarah K. Volkman, Matthias Marti, Daniel E. Neafsey, Nicanor Obaldia III.

**Data curation:** Lucas E. Buyon, Ana Maria Santamaria, Angela M. Early, Jose Lasso, Mario Avila, Daniel E. Neafsey, Nicanor Obaldia III.

**Formal analysis:** Lucas E. Buyon, Angela M. Early, Daniel E. Neafsey, Nicanor Obaldia III.

**Funding acquisition:** Daniel E. Neafsey, Nicanor Obaldia III.

**Investigation:** Ana Maria Santamaria, Mario Quijada, Itza Barahona, Jose Lasso, Mario Avila, Daniel E. Neafsey, Nicanor Obaldia III.

**Methodology:** Lucas E. Buyon, Angela M. Early, Daniel E. Neafsey, Nicanor Obaldia III.

**Project administration:** Sarah K. Volkman, Matthias Marti, Daniel E. Neafsey, Nicanor Obaldia III.

**Visualization:** Lucas E. Buyon, Daniel E. Neafsey.

**Writing – original draft:** Lucas E. Buyon, Daniel E. Neafsey, Nicanor Obaldia III.

**Writing – review & editing:** Lucas E. Buyon, Ana Maria Santamaria, Angela M. Early, Mario Quijada, Itza Barahona, Mario Avila, Sarah K. Volkman, Matthias Marti, Daniel E. Neafsey, Nicanor Obaldia III.

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
