## [Decision Letter · Decision Letter 0]

28 Sep 2020

Dear Mr Buyon,

Thank you very much for submitting your manuscript "Population Genomics of Plasmodium vivax in Panama to Assess the Risk of Case Importation on Malaria Elimination" for consideration at PLOS Neglected Tropical Diseases. As with all papers reviewed by the journal, your manuscript was reviewed by members of the editorial board and by several independent reviewers. The reviewers appreciated the attention to an important topic. Based on the reviews, we are likely to accept this manuscript for publication, providing that you modify the manuscript according to the review recommendations. 

Sincerely,

Margaret A Phillips, Ph.D.

Deputy Editor

Margaret Phillips

Deputy Editor

Reviewer's Responses to Questions

**Key Review Criteria Required for Acceptance?**

**Methods**

-Are the objectives of the study clearly articulated with a clear testable hypothesis stated?

-Is the study design appropriate to address the stated objectives?

-Is the population clearly described and appropriate for the hypothesis being tested?

-Is the sample size sufficient to ensure adequate power to address the hypothesis being tested?

-Were correct statistical analysis used to support conclusions?

-Are there concerns about ethical or regulatory requirements being met?

Reviewer #1: (No Response)

Reviewer #2: The methods are well-suited for the questions addressed and the analyses well-conducted.

Reviewer #3: -Are the objectives of the study clearly articulated with a clear testable hypothesis stated? Yes

-Is the study design appropriate to address the stated objectives? Yes

-Is the population clearly described and appropriate for the hypothesis being tested? Yes

-Is the sample size sufficient to ensure adequate power to address the hypothesis being tested? Unable to assess

-Were correct statistical analysis used to support conclusions? Yes

-Are there concerns about ethical or regulatory requirements being met? No

Abstract and Author’s Summary

Line 35: “spanning the entire period of the collection (2007-2019)” and lines 56-57 “we collected 100 Panamanian P. vivax samples from two collection periods (2007- 2009 and 2017-2019).” 

o Can the authors clarify whether the samples were collected throughout the 2007-2019 period as indicated in the Abstract, or only in the periods 2007-2009 and 2017-2019 as indicated in the Author’s Summary? If the latter, it should probably be more specific in the introduction, as the reader might think this is continuous longitudinal sampling over a decade. 

Methods

Lines 344 – 346 “We extracted DNA from the filter paper blood spots using the Chelex method as described (16) for samples obtained during 2007-2009 and with the Qiagen® DNA mini kit for samples obtained during 2017-2019.” 

o Would there be any expected impact on results due to different in extraction method? Is there a need to account for that potential in data analysis methods?

Lines 409 -410: “to ascertain recent common ancestry among Panamanian and Colombian samples collected previously from the global P. vivax population study”. 

o Some sentences briefly giving some background to the global P.vivax population study would help the reader less familiar with the area.

**Results**

-Does the analysis presented match the analysis plan?

-Are the results clearly and completely presented?

-Are the figures (Tables, Images) of sufficient quality for clarity?

Reviewer #1: (No Response)

Reviewer #2: Fine, although I would have liked to see a display of the pairwise IBD.

Reviewer #3: -Does the analysis presented match the analysis plan? Yes

-Are the results clearly and completely presented? Some clarifications needed regarding proportion of samples excluded with low genetic distance from CL1.

-Are the figures (Tables, Images) of sufficient quality for clarity? Yes

Results

Lines 156 – 157: “We observed that as the average proportion of high coverage sites for sample pairs 157 increased, pairwise IBD estimates correspondingly increased as well.” 

o Could the authors include briefly some of the factors that lead to low proportion of high coverage sites for samples? Is the distribution of the lower sample quality related to the sample collection periods, field collection/storage or extraction/sample processing methods that might introduce a systematic bias in the analysis?

Lines 182-183: “the majority of excluded samples in both collection periods (17/21 and 4/16 for the 2007-2009 and 2017-2019 collection periods respectively) exhibited very low levels (0-1%) of genetic distance with CL1 samples. Seven of the 2017-2019 samples exhibited 0-1% genetic distance with CL1 also showed 0-1% 186 genetic distance with CL2, making the identity of these samples uncertain.” 

o This means that the majority of the excluded samples e.g., of the 21 excluded samples, 17 had low levels of genetic distance from CL1, but only 4 of the 16 excluded samples in the later collection period had low genetic distance? However, the next sentence indicates that 7 samples from the latter collection period had low CL1 genetic distance. Does this mean more specifically that the majority of the samples collected in the earlier period were those that showed low genetic distance, but less than half of those in the latter collection period did? Could the authors clarify? Does this indicate anything about bias introduced by sample collection or processing, as per comment above?

**Conclusions**

-Are the conclusions supported by the data presented?

-Are the limitations of analysis clearly described?

-Do the authors discuss how these data can be helpful to advance our understanding of the topic under study?

-Is public health relevance addressed?

Reviewer #1: A recent study using microsatellites (https://journals.plos.org/plosntds/article?id=10.1371/journal.pntd.0008072) and not so recent one (https://journals.plos.org/plosntds/article?id=10.1371/journal.pntd.0008072), as well as WGS data (https://www.nature.com/articles/ng.3588) point to a P. vivax population in Central America and Mexico that it is clearly separated from South America. I am surprised these studies as not mentioned and discussed in light of the new data.

Further, on lines 241-246, I am puzzled that the authors decided to discuss similar patterns observed for P. falciparum, but not P. vivax. Clonal or near-clonal P. vivax populations have been described before, e.g. here https://www.nature.com/articles/s41467-018-04965-4. While P. falciparum can be discussed, the evidence we have on P. vivax should not be omitted. Overall this part could be referenced better.

Reviewer #2: The discussion of the ability of genome sequence data to support inference of imported cases should be expanded (see below).

Reviewer #3: -Are the conclusions supported by the data presented? Yes

-Are the limitations of analysis clearly described? Some description of the factors that might lead to the limitations highlighted by the authors would be useful.

-Do the authors discuss how these data can be helpful to advance our understanding of the topic under study? Yes

-Is public health relevance addressed? Yes

Line 261: “variable malaria transmission”

o Do the authors mean to say spatially heterogeneous or focal transmission (as opposed to temporally variable)?

**Editorial and Data Presentation Modifications?**

Reviewer #1: (No Response)

Reviewer #2: (No Response)

Reviewer #3: Abstract and Author’s Summary

Lines 53-55: “Understanding parasite transmission patterns and identifying imported cases is critical to help Panama and other countries succeed in their elimination efforts. Genomic epidemiology and population genomics can help provide information needed to inform malaria control policy.” 

o If word count allows, these sentences could elevate the intro of the paper by being a bit more specific in how the data could be used to guide elimination efforts or inform control policy. This could potentially have a Pv focus, which is why this research is unique and should be highlighted. E.g., how might “differentiation between imported infections or sustained local infection as the underlying cause of persistent P. vivax incidence” as mentioned in the introduction lead to policy decisions.

Introduction

Lines 116-122: “We found the vast majority of P. vivax cases in Panama belong to a single highly related lineage that has persisted for at least a decade. Furthermore, we observed a second smaller clonal lineage concentrated near the Panamanian-Colombian border. We also found several samples that shared no relatedness with any other sample, which may represent either localized pockets of outbred P. vivax transmission or imported cases. These findings suggest the Panamanian P. vivax population has extremely low genetic diversity and is on the cusp of elimination” 

o These seem to be results, so should be left out of the introduction for now or moved to the results/conclusions section, if not already summarised there.

Materials and Methods

Should this section be moved up to come after the Introduction and before the Results? Check with PLoS formatting guidelines

Discussion

Line 261: “variable malaria transmission”

o Do the authors mean to say spatially heterogeneous or focal transmission (as opposed to temporally variable)?

**Summary and General Comments**

Reviewer #1: This study presents interesting data, showing very strong clonality of P. vivax in Panama, with some lineages persisting for over 10 years. It is mostly well written and I do not have any major concerns, but there are a few points that are misleading and need to be corrected. 

Further, I feel the discussion misses the opportunity to better discuss the new data in light of previous studies pointing to a genetically unique Central American P. vivax population, and similar clonal population structure from other parts of the world (I wonder why P. falciparum is discussed instead).

Minor comments

Line 41: It is difficult to understand what is meant by “The small number of genetically unique …”. It would be clearer to mention CL1 and CL2 in lines 34-37, and then on line 41 to clarify that CL2 is referred to.

Line 72: I believe it is now established that P. ovale wallikeri and P. ovale curtisi are 2 different species.

Line 77-78: “However, recent studies suggest this parasite species causes a significant global health burden (1,2).” 

Reference 2 is from 2007, I would not call that ‘recent’

Line 79: P. ovale also forms hypnozoites, thus they are not unique to Pv

Line 79: It is very rare that Pv relapses years after the initial infection, most relapses occur much earlier. Please provide a more realistic statement.

Define IBD at first use

Define VCF at first use. I also wonder why the authors do not use the term ‘data set’ or similar instead of repeatedly refer to ‘VCF’ (i.e. the file format). 

Lines 178-188: When the analysis was restricted to 264 SNPs, did any results change for the samples already analyzed with the WGS data? Did some samples no longer cluster with CL1 or CL2? It is only described how the additional samples cluster, but not whether any results had changed if all samples had been typed with only 264 SNPs.

Line 212: The title “Genomic data both support and refute travel history data” is misleading and leaves the reader wonder what is going on. Stating “Genomic data matches travel history in three out of four cases” would be much clearer.

Line 242: Reference 30 is a study on P. vivax, not P. falciparum.

Lines 310-311: “While case importation remains a threat, the lack of evidence of outcrossing suggests it may not be sufficient to prevent elimination under present circumstances.”

This sentence is confusing. Should “outcrossing” be replaced with “onward transmission”? I don’t see why outcrossing per se is a requirement for imported infections to be a thread for elimination. 

Line 313: It appears a word is missing before ‘reversal’, likely it should read ‘prevent reversal’.

Reviewer #2: The study by Buyon et al. describes 59 genome sequences from Panama and their analyses with respect to the global and continental diversity of P. vivax. The data are interesting and the analyses, for the most part, well conducted and convincing. I have some minor comments and suggestions on the presentation of the analyses, but the manuscript is overall well written and interesting. My main comment is to expand the discussion of how genetic information can (or cannot) inform on case importation based on the region genetic diversity and the specificities of P. vivax biology.

Main comment:

I think the authors missed an opportunity to discuss the ability to identify imported cases using genetic data (which would better reflect the title of the paper). Based on their data and analyses, it is not clear that genomic information provides sufficient information to rigorously identify the origin of a parasite: the differences between continents is clear and convincing but it is not obvious that one can discriminate with high confidence clones from Brazil vs. Peru, or Mexico vs. Nicaragua. Maybe the authors can provide some statistical support for such analyses and discuss the limitations for malaria control (e.g. identifying robustly imported cases and/or the origin to these cases). The high clonality of Panama is clearly an advantage in this case and this could be also discussed (as the prevalence goes down, it might be easier to identify imported cases). Finally, the role of dormant P. vivax parasites should be at least mentioned as a potential complication when trying to reconcile recent travel history and parasite genetic diversity. 

Minor comments and editorial suggestions:

Complexity of infection. The sequence coverage for many samples is very low (especially for the 2017-2019 samples). This limits the ability of the authors to detect polyclonal infections (if any) and this limitation should be at least noted. The COI (or lack thereof) could also be used in the discussion as further support for the statement that the P. vivax population in Panama has low genetic diversity and is highly clonal.

IBD analysis. Instead of Fig2B (that could be moved to supplemental), I would have like to see the pairwise IBD among samples from Panama (e.g., a heatmap of IBD), with maybe the Colombian samples to provide a comparison point. This would strengthen the point the authors made and emphasize the uniqueness of the P. vivax population in Panama.

Figure 4. It would be great if the authors could use the same colors and symbols for A and B

Line 29: emphasizing the “96 samples” in the abstract is a little misleading and the authors should replace by the 59 actually successfully sequenced and analyzed.

Line 139: please indicate the number of Colombian samples included

Line 171-174: Duplicated information. Remove the first sentence of the paragraph?

Line 198: replace “zero” by “little”

Line 234: I am not sure to understand correctly the statement of “extremely low clonal diversity”… Replace by “extremely low genetic diversity” or “high clonality”

David Serre

Reviewer #3: This study provides a useful analysis, describing parasite population dynamics that may guide elimination strategies in Panama. The authors highlight a number of ways in the conclusions in which the data can inform policy - this could be elevated by briefly describing follow-up studies or programmatic actions to illustrate the translation of genomic data into real-time epidemiological application. Is genomic data ready for immediate inclusion in case investigation and response, or does it need to be paired with other data and surveillance in order to be actionable. If so, what are these things factors that need to be considered.

PLOS authors have the option to publish the peer review history of their article (what does this mean?). If published, this will include your full peer review and any attached files.

Reviewer #1: Yes: Cristian Koepfli

Reviewer #2: Yes: David Serre

Reviewer #3: Yes: Lindsey Wu
---

## [Editor Report · Decision Letter 1]

6 Nov 2020

Dear Mr Buyon,

We are pleased to inform you that your manuscript 'Population Genomics of Plasmodium vivax in Panama to Assess the Risk of Case Importation on Malaria Elimination' has been provisionally accepted for publication in PLOS Neglected Tropical Diseases.

Best regards,

Margaret A Phillips, Ph.D.

Deputy Editor

Margaret Phillips

Deputy Editor

---

## [Editor Report · Acceptance letter]

2 Dec 2020

Dear Mr Buyon,

We are delighted to inform you that your manuscript, "Population Genomics of *Plasmodium vivax* in Panama to Assess the Risk of Case Importation on Malaria Elimination," has been formally accepted for publication in PLOS Neglected Tropical Diseases.

Best regards,

Shaden Kamhawi

co-Editor-in-Chief

Paul Brindley

co-Editor-in-Chief
